# The Need for Shari'ah-Compliant Awqāf Banks

**Hanan Gabil [1], Benaouda Bensaid [2], Tahar Tayachi [1],\* and Faleel Jamaldeen [1]**

[1] Finance Department, College of Business, Effat University, Jeddah 22332, Saudi Arabia; hhgabil@effatuniversity.edu.sa (H.G.); jfaleel@effatuniversity.edu.sa (F.J.)
[2] Faculty of Islamic Sciences, Istanbul Sabahettin University, 34303 Kucucekmece Istanbul, Turkey; benaouda.bensaid@izu.edu.tr
\* Correspondence: ttayachi@effatuniversity.edu.sa

**Abstract:** Bridging global economic inequalities calls for effective financial alternatives such as awqāf banks to better attend to the needs of the poor and underprivileged. This is expected to address the root causes of poverty and ensuing economic gaps, improving much of the living standards whether pertaining to education, health, shelter, employment or basic social services while reducing the state's economic and financial burden. We envision awqāf banks as institutions which are established through cash awqāf and which operate multiple awqāf funds alongside an assortment of financial instruments. The main use of their awqāf funds are the issue of low-cost credit to the poor, economically disadvantaged and underprivileged, instead of focusing solely on generating and maximizing shareholder profits. This is to support the economy through of steady and sustainable growth, effectively raising the lower bar on per capita income and lifting multitudes out of poverty and need. This paper explores how low-cost credit can be provided to the poor or lower income demographics through awqāf banks, while addressing relevant issues such as Shari'ah compliance, services rendering, investment and awqāf distribution. This paper also examines current studies on awqāf in relation to finance and banking, the basic functions, and characteristics of the Shari'ah-compliant awqāf bank, as well as evaluations of awqāf banks. Current studies show that there is a legitimate need for Shari'ah-compliant awqāf banks which not only providing services for its beneficiaries but also manage investments and awqāf funds that contribute to overall national development and economic growth. This study would be of high relevance to experts, practitioners, financial managers, regulators, and policy makers in the fields of awqāf, banking and finance.

**Keywords:** awqāf bank; cash awqāf; awqāf management; Islamic Endowment

## 1. Introduction

Poverty has long been a major issue of concern among developing and underdeveloped countries. Poverty is defined as a situation in which individuals are unable to meet the bare subsistence such as food, clothing, and shelter, along with the lack of employment, skills, assets and self-esteem, with restricted access to social and economic infrastructure that would ensure maintaining a minimum standard of living (Okpara 2010). There are several plausible explanations of the prevalence of poverty such as inequality of resource ownership which leads to unequal distribution of income, thus causing poverty or differences in the quality of human resources, which also leads to poverty that may result from differing access to social capital (Setiawan 2016). Poverty has been the greatest challenge of all time. Therefore, in order to alleviate it, governments across the globe have come up with the idea of financing the poor through banks (Adeyemi 2008). However, the poor tend to have circumscribed access to banks' services because of the lack of physical collateral and the cumbersome process of transactions which significantly discourage those with less education to approach banks

(Imai and Azam 2012). This means that to make this idea work, the provision of financial services must be universal, covering all people as shown in the empirical evidence which suggests that an inclusive financial system significantly improves growth, alleviates poverty and provides wider economic opportunities (Raman 2012).

Several studies have been conducted on the relationship between financial development and growth (Chistopoulos and Tsionas 2004). The question of whether financial development significantly contributes to poverty reduction persists (Honohan 2004). A strong correlation was found between economic growth and poverty alleviation in developing countries (Knowles 2005). The ability of the economy to produce jobs can significantly affect the relationship between economic growth and poverty. Moreover, the sound fiscal policies and social protection can prove beneficial to poverty reduction (Paci et al. 2004). Mallick's study (Mallick 2008) suggested that the availability of credit for businesses must be the key component of monetary policy to combat poverty as a credit provision, especially for agriculture which reduces poverty significantly. However, the high cost unit of small loans is also an impediment for poor households to access bank credits because initial set-up cost is being required by financial intermediations offers, of which poor households cannot afford (Greenwood and Jovanovic 1990). Poor households are not able to risk their savings and fall further below in income distribution, which is why international aid is greatly involved in micro-financing institutions, which is inherently problematic (Jeanneney and Kpodar 2011).

Despite literature revealing the positive impacts of financial development on alleviating poverty, income inequality has been on the rise in most countries since the early 1980s (Seven and Coskun 2016). One of the plausible claims is that there is no universal way to measure or evaluate whether provision of credit is meeting its social goals or not. As a result, banks only focus on easily measured financial outcomes which do not depict the true picture of poverty alleviation (Copestake et al. 2005). This is because if banks measure the success rate of loans repayment, they then ignore the fact related to how those loans are benefitting poor people socially.

In particular, when the poor are still hungry, there are rare chances that they will use micro-finance for investing in business, which implies that provision of credit by banks is causing more harm than good due to the fact that accumulated debt would lead already poor individuals into further destitution and an unending cycle of debt. Secondly, financial development can only be effective in the alleviation of poverty when banks reach out to the poorest of the poor. For instance, a study conducted on the Vietnam Bank for Social Policies (VBSP) to investigate whether the institution is actually reaching out to the poor or not, shows that only 12% of poor households participated in the program while up to 67% of the participants belonged to non-poor households (Cuong 2008). Thirdly, provision of micro-credit may lead to polarization between the poor and may even be detrimental for the poorest of the poor in view of the fact that many institutions require individuals to guarantee the return of the full loan amount. This means that as the size of the loan grows, it becomes harder to repay it (Copestake 2002).

Moreover, despite the extensive bank networks and systems of today, poverty still persists. More than 500 million continue to live in urban slums with close to one billion people in persistent hunger in all regions of the Muslim world (Ahmed 2004, p. 150). Banks did only a little to bridge the global socio-economic and technological gap; their micro-financing of privileged segments is set for profit-making at the expense of the poor and languishing (Abu Zahrah 1971, pp. 156–67). Developed countries are doing a shady job of development in developing countries; and, when countries in the third world are largely funded by non-governmental organizations, world banks and international monetary funds that make most of them incur unfathomable amounts of debt, causing more damage to already prevailing conditions of poverty in those countries (Ahmed 2004, p. 150).

The Islamic tradition is replete with moral, ethical, and legal teaching on wealth generation, development, and distribution. The Qur'an calls for equitable distribution of wealth and resources. Poverty should be eradicated as showcased in the numerous injunctions of voluntary sharing of resources with the poor and needy. This highlights Muslim responsibility on poverty alleviation, support, and integration of the underprivileged into economic and financial

empowerment (Young 2010, pp. 201–23). Justice, compassion, and solidarity towards the poor eases the socio-economic empowerment of the poor and needy. This requires the state, community, and people to commit together to alleviating poverty and helping those without food or healthcare (Dawwabah 2005, pp. 48–75). The Qur'an (59:7) acknowledges economic disparities among rich and poor and sets the vision, measures, and incentives of its alleviation. The Qur'an, for instance, encourages Muslims to strive towards spiritual and economic empowerment while promoting acquisition and development of wealth. Dependence on others is seen as a form of self-caused demise (Renneboog et al. 2008, pp. 1723–42). The idea is that one should commit to hard work, diligence, and transparency for wealth search with only a few limitations on how to look for it.

According to the Qur'an, resources should be explored (Buttle 2007, pp. 1076–88), used and disposed of consistently with justice. In their effort to counter unequal distribution of wealth, both the state and society need to share the responsibility of protecting and providing for the poor through moderate consumption of resources, as well as to assure provision for the under-privileged. The shared responsibility of the state and the community is to ensure effective distribution of wealth. The duty of economic development that is incumbent on the state demands distribution of wealth in accordance with the stipulations of Shari'ah (Siddiqi 2004, p. 81). The state should initiate programs that support and empower the poor by way of the provision of sustainable employment opportunities. This requires balancing out the rights of the poor to treasury allocation without neglecting the remaining population; others are owed a spiritual and ethical duty of care (Hassan and Shahid 2010, pp. 309–28). Morally speaking, there is a close relationship between the individual and community in the Muslim society (Grodach 2011, pp. 300–9). Still, a similar relationship exists between economic and spiritual empowerment and development.

Speaking of the context of economic transactions, God created resources in abundance (Dunya 2002, pp. 57–82). The Qur'an holds high regard for those who sacrifice their wealth for the poor by way of charity (*sadaqat*) or *awqāf* (Zaman 1999, pp. 1–8). It is mandatory to give zakat and recommended charities (*sadaqat*) to those in need or less privileged (Abdel Mohsin 2009, pp. 17–27). Those go in conjunction with cooperating with one another. One needs to be righteous and uphold the principles of fairness while keeping away from fraud and unlawful betting. Individual obligations towards others are reinforced when there is freedom to act in total disregard of self-interest of material wealth and personal well-being (Zaman 1999, pp. 1–8). In the quest to disregard self-interest, Islamic law appreciates when those are used in the provision to the economic and spiritual advantage of less privileged people.

Spirituality cuts across both, striving to sustain the self in the best way possible and ensuring that poor neighbors are supported to attain self-sustenance. This is essential for socio-economic development since the individuals providing help to neighbors do not look forward to reward or appreciation for what they do (Klugman 2009, pp. 13–46). Philanthropy echoes a sense of responsibility and makes one consume that which is adequate without wasting resources, especially when others are hungry. The resources God created should not be exhausted excessively at the expense of others who lack them (Kahf 1998, pp. 2–4). When hard work, labor indulgence and investments are combined with the principle of altruism, then the poor will not be exploited by those with illicit and malicious intentions of domination or economic exploitation (Kahf 2008, pp. 2–4). On the contrary, the poor will be increasingly exposed to opportunities of development, whether spiritual or socio-economic, and will have fairer chances of growth, empowerment, development, and integration.

Let us, before proceeding further, turn to the fundamental concept of awqāf. Awqāf is an Islamic religious endowment fundamentally set to freeze the proprietorship of assets, as is the case of a voluntary and irrevocable dedication of one's wealth or a portion of it—in cash or kind (such as a house or a garden), and its disbursement for Shari'ah-compliant projects such as mosques or religious schools. Awqāf helps eradicate poverty while ensuring sustainable support of the poor (Mishra 2006, pp. 1538–45). The sustainable nature of awqāf finds support in the Qur'an, tradition of the Prophet Muhammad, and Islamic law. Prophet Muhammad is reported to have said that deeds

of individuals cease to continue upon death except for those dedicated as charity (*sadaqa*), beneficial knowledge or pious children. Awqāf is also characterized with principles like inalienability of the rights of the poor, permanent endurance, and potential to generate income. However, there is a need to gear awqāf towards making income generation and re-generation. More light should be shed on this issue, which can be fulfilled in many different ways (Haji Mohammad 2015, pp. 37–73).

It follows that responsible resource management proves the care of Muslims not only about their present generation, but also of their posterity (Dreher 2006, pp. 769–88). Prophet Muhammad established what is known as *habs* (endowment) which is economically self-reliant, charitable, and sustainable. Unlike other Islamic institutions such as charity (sadaqa) and religious donations or compensations (*kaffarah*), awqāf remains perpetually charitable (Obaidullah 2007, p. 3) because it is not subject to revocation or transfer. Awqāf can also generate its own income and fund charitable activities of the poor and underprivileged. Awqāf institutions can be presented in different ways, including cash and real estate (Barizah Abu Bakar et al. 2005).

Earlier literature on the financial dimensions of awqāf institutions has focused largely on the way the religious nature of awqāf resulted in successful financial development of awqāf properties and related initiated investments are initiated (Siddiqi 2004, p. 81). In the 19th Century and after, many political conflicts in the Muslim nations caused serious deterrence for awqāf. Cash awqāf was not very predominant and the real estate awqāf took all the limelight (Barizah Abu Bakar et al. 2005). Issues of profit-making properties remain generally lacking. Only a few have delved into issues of funding of awqāf institutions, portraying it as a lending and benevolent institution. Most characterize awqāf as an institution set to counter the influence of the conventional financial institutions but which had inadequate assets to enable it to overcome the challenges that it was due to face (Amin et al. 2003, pp. 59–82). There exist, however, several avenues to address such an information deficit, including, for example, the development of original awqāf practice of socio-economic empowerment by way of improving maximum wealth creation (Mohammad 2011a, p. 381). This would ensure that awqāf funds, or other funds collected by the bank, are used for obtaining additional property and funds, which would be used again for favorable micro-financing programs for less privileged people who cannot afford the bank charges on financial lending.

Despite the growing literature on the awqāf, finance and the economy, and on how the awqāf impacts socio-economic development, the discussion of the establishment of a awqāf bank that would effectively manage awqāf capital while microfinancing the poor segment and contributing to the overall development of society still needs more focus and attention. This is crucial as it places the awqāf discourse on a much more practical platform, touching not only upon the different aspects and experiences of financing, banking, and development, but also upon the very criteria of a Shari'ah-compliant awqāf bank today.

This research discusses the rationale, vision, and relevance of a Shari'ah-compliant awqāf bank, in addition to relevant issues including the legality of a cash awqāf (*awqāf naqdi*). This study first provides some basic definitions of the awqāf and social banks. It also addresses questions of bank management and structures such as Shari'ah compliance, service rendering, investment and awqāf distribution, as well as how to provide interest-free and affordable loans to the poor and lower income demographics and empowerment of beneficiaries in need, poverty eradication, removing socio-economic inequalities while honoring the conditions and stipulations of endowers (*waqifs*) and contributing to overall national development and economic growth.

The current research is important not only to the theoretical field of awqāf studies, but also to new fields and issues of awqāf, banking and finance, digitization, greening, and, more importantly, to the core impact of awqāf on bettering human lives and the world. Much of the discussion around the Shari'ah awqāf bank touches on areas of fiqh innovation, creativity, and ijtihad. However, it would remain critical to advancing our understanding of the ways and means of helping the poor, needy and underprivileged, and also of the sustainable strategies that would secure and develop the awqāf capita while changing the lives of people and their environments for the better.

## 2. Awqāf Institutions and Social Banks

Institution of awqāf can be described as charity profit-making organizations with great potential power of community empowerment and nation building. They may appear as privately-owned financial institutions, social banking institutions, income-generating institutions, and institutions interested in charity activities and the redistribution of wealth (Ahmed 2004, p. 150). Moreover, they have various objectives and respective challenges such as being under-financed or being short-lived because of the unfavorable conditions of poverty without financial aid from other organizations (Arnawut 2005, pp. 3–20). Social banks, on the other hand, represent non-profit organizations dedicated to socio-economic welfare. Islamic social banks seek to alleviate poverty while contributing to the community's economic development and upbringing of micro-finance institutions against poverty (Ammar 2006, pp. 11–19).

In the context of the Muslim world, increased attention is given to poverty alleviation. There are also ongoing evaluations of the contributions and effects of awqāf on the sustainability of socio-economic development. However, the challenge of how to systematically eradicate poverty persists. This, however, requires investment in all possible anti-poverty financial measures, and more importantly, perhaps in establishment of the awqāf bank, with characteristics that are acceptable to the culture and religious affiliations of the community alongside checking the ways according to which awqāf would achieve its goals effectively (Cizakca 2004, pp. 40–59). Unlike conventional banks and indigenous awqāf financial institutions, however, awqāf institutions provide an alternative financial structure as shown Figure 1 and amalgamates and also consolidates the properties of both institutions and sets them in one single package (Vakulabharanam 2005, pp. 971–97). This is to be held in the best interests of all parties in the Muslim community; whether un-bankable or more privileged, which as Mohammad (2015, pp. 37–73). Noted, renders it in the best interest of the poor.

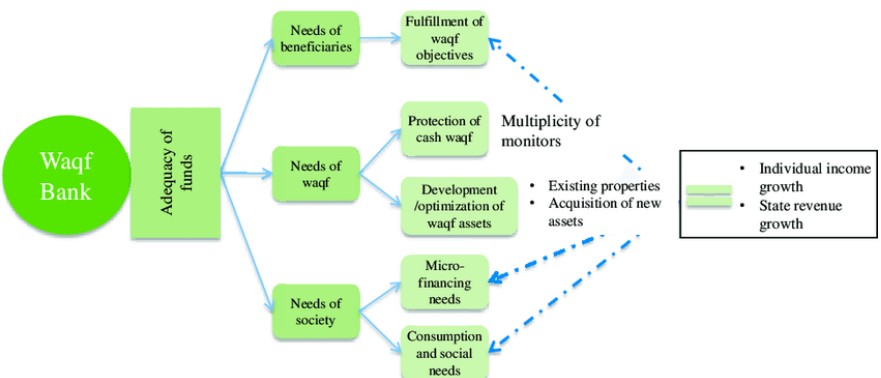

**Figure 1.** Waqf Bank and its Utility to waqf and society (source: Mohammad 2015).

According to the current financial model, income generating operations largely target wealthy and privileged segments and do not embrace micro-financing models in countries like Yemen, Bangladesh, and England inter alia (Ammar 2006, pp. 11–19). The model of finance applied in those countries tends to be more conventional and the few current awqāf social financial institutions major on profit making initiatives to finance their charitable activities. In countries like Bangladesh for instance, only a part of the bank deals with awqāf (Young 2010, pp. 201–23). Still, that part does not use the gained funds to supply awqāf-based charitable activities; on the opposite, it provides the bank with financial funding, which fails to extend any financial help to the poor; and as profit-making institution, it is only concerned with profit maximization, however, in the name of a social awqāf institution, as is the case of the Islamic Social Bank in Bangladesh or the Vakif Bank in Turkey (Zarqa 1994, pp. 55–62).

With regards to the Islamic social banks making profits out of earnings of the services offered by the bank, which are set to finance social welfare activities and especially micro-financing, indigenous awqāf financing institutions are based on pure charity or organizations with no prospect of making

profits. They are also conducted according to awqāf fundamental principles, rules, and values and would resemble other financial institutions in need of funding, however, remain different from conventional banks. Their lending would be advantageous to individuals, parties, or institutions, Muslims, and others alike. They would also be different in many aspects of social banking systems (Abu Zahrah 1971, pp. 156–67).

In spite of the several inherent yet fundamental distinctions, there exists, in England, for example, banks that masquerade as awqāf institutions, with their own forms of operation through which they would obtain and use funds to provide interest-based loans for the poor and underprivileged that would otherwise lack collaterals and securities to make them eligible for loans. Their particular feature however, is that the parties providing funds must not be parties interested in achieving charitable objectives, but, on the contrary, parties which are after a conventional means to an end (Zarqa 1994, pp. 55–62). The probable notion is that the bank is generally any other conventional bank. The bank, in turn, uses their earnings from those financial relationships to finance their charitable activities; that is, provision of unsecured loans to the poor and unprivileged in the Muslim communities, as is the case with the Charity Bank of England and the England's Industrial Common Ownership Finance Ltd (ICOF). Their focus is on social banking and activities pertaining to the poor in a bid to bridge the large gap among rich and poor in the Muslim community. Those very activities have been influenced by the awqāf incessant and persistent bid to alleviate poverty in most Muslim countries.

However, in view of the inherent inter-connection of awqāf with the Muslim belief, spirituality, morality, and law, one would assume that awqāf operations would only be appealing in religious and faith-based contexts. Having said that, one should note that there exists still a gross disparity in the execution of awqāf initiatives related in most to the fact that the effectiveness of awqāf operations is largely connected with the development index and socio-economic levels, as it would be much easier, for instance, to implement awqāf resolutions, applications or solutions in countries like the United Arab Emirates and Kuwait, and extremely difficult in others like India, Pakistan or Iraq. In countries like Somalia, for instance, economic disparity is predominant, especially with the ongoing political conflicts and instability, making it impossible to implement the initiatives of awqāf financial institutions (Mohammad 2011b, pp. 250–54).

## 3. Cash Awqāf in Islamic Law

Awqāf banks are the next step up from cash-based Awqāf, the latter being a natural consideration once the cash system is acknowledged by a Muslim community (Arnawut 2005, pp. 3–20). Awqāf bank are subject to Islamic law, which covers various issues such as capital generation, management and even ownership. However, awqāf represent a controversial issue among Muslim scholars despite potentially being one of the best modes of sustainable profit generation. This reluctance is largely because the idea of a cash-based awqāf is not explicitly addressed cash awqāf in the primary sources of Islamic law (Renneboog et al. 2008, pp. 1723–42). Some jurists have considered it permissible through legal analogy as is the case with many other legal issues. Some have accepted it, by considering cash for cash awqāf as a movable property in that early Qur'anic notions were largely more concerned with property than a medium of trade (Roodman and Morduch 2014, pp. 583–604).

More generally, there is no one school of jurisprudence whose jurists agreed unanimously on the prohibition of cash awqāf. The schools with the strongest legal support for cash awqāf are the Maliki and the Hanafis schools. The Hanbalite school adopted a similar stance as shown in the ruling made by its leading jurists Ibn Taymiyyah. The Shafi'i school is perhaps the least favorite in the issue of cash awqāf (Dunya 2002). Abu Su'ud wrote a treatise on cash awqāf in which he supported the opinion of permissibility based on the principle of juristic preference (*istihsan*), transactions of people, and customs. He issued seventy-nine fatwas in favor of cash awqāf investment through Islamic equity-based partnership contract (*mudharabah*) (Abu al-Suud 1997, p. 13). Nevertheless, most Muslim scholars opt for the permissibility of cash awqāf, viewing it as a means of trade and comparable asset (Mandaville 1979). Their argument is based on the recognition of both the Qur'an and hadith of

moveable property. Few jurists however, recognized cash awqāf to procure a loan. Others recognize it in the light of their consideration of awqāf. However, some views in the Shafiʻi school of law rejects cash awqāf based on their own legal methodology and juristic interpretations. The validity of the cash awqāf, however, is still accepted, especially when it sustains and derives its own profits and is operated to procure loans and establish investments. For Mandaville, many other schools began to recognize the validity of cash Awqāf, including the Hanafis and the Malikis who hold different justifications for recognizing cash awqāf, such as its ability to influence the establishment of investments (1979).

Contemporary jurists rule in favor of the admissibility and acceptance of cash awqāf for its purposes in investments. They maintain permissibility of cash awqāf in investments of bonds and banks. Some go further to recognize not only currencies, but also liquid assets pertaining to cash awqāf. In 2004, the Islamic Fiqh Academy of the Organization of the Islamic Conference (OIC) authorized the perpetual and temporary cash awqāf in the resolution No 140 (15/6) on investment in awqāf and its yields and rents. The Accounting and Auditing Organization for Islamic Financial Institutions (AAOIFI) authorized the cash awqāf, stocks, funds and sukuk awqāf in Sharia Standard No 60 (amended), items No. 2/4/13, 2/4/14 and 2/4/15 (The Accounting and Auditing Organization for Islamic Financial Institutions 2019).

According to contemporary Muslim jurists, cash awqāf has numerous advantages including the creation of more opportunities for socio-economic empowerment. It has a very high potential of constantly sustained income generation and regeneration (Arnawut 2005, pp. 3–20). Cash awqāf has many elements that qualify it to play a significant role in the fulfillment of both the charitable and development mission of awqāf in the most satisfactory fashion. Attention and efforts should then be given to this type of awqāf for its formulation and development as it represents a crucial entry in the rejuvenation of awqāf in our modern times (Ahmed 2004). Cash awqāf is far more important since it is more productive compared to land, buildings, books, cattle and so on, as it is testified by the existing research studies and have been found profitable in the practices of modern Islamic financial system (Ab Aziz et al. 2013).

Cash awqāf also appears to be flexible; facilitative for donations and realizes liquidity in most of the time (Mahadi 2015, p. 70). The investments initiated through cash awqāf can be diversified and used in many opportunities, demonstrating that it stands out as the most effective method for joint awqāf creation. Furthermore, the fact that the core assets are of a liquid nature also makes it beneficial in all sorts of economic transactions, including income generating activities for the awqāf which would otherwise not be feasible (Barizah Abu Bakar et al. 2005). A simple example is the facilitation of operational capital which can most efficiently be implemented by having liquid assets.

Cash awqāf has been proposed as one of the most effective modes to eradicate poverty in the Muslim community (Grodach 2011, pp. 300–9). However, this is subject to many limitations, including the deficiency of awqāf fund management. In addition, some of the generated funds might be abused by awqāf managers (*nuzzar*) for fraudulent uses, and hence facilitate the destruction of cash awqāf institution. In many scenarios and as a practical example, the funds gained and those earned through awqāf are not usually accounted for, and so therefore most parties usually use those for personal benefits instead of charity purposes (Zaman 1999, pp. 1–8). The remedy for awqāf abuse problems lies in the creation of awqāf bank, which would provide various services for income generation, and which again are used to procure services to the poor and needy. One needs to keep in mind, however, that the true essence of the creation of awqāf is to facilitate the management of awqāf funds and create chances for investments in various avenues (Buttle 2007, pp. 1076–88).

Pertiwi et al. (2019) discussed three types of awqāf management: traditional, semi-professional and professional awqāf management. This latter relates to the current discussion of cash bank awqāf. The traditional awqāf management is characterized by the placement of awqāf as worship or rituals, set in awqāf property in the form of physical development, such as mosques, boarding schools or burial grounds. Semi-professional awqāf management is characterized by the development of awqāf assets. The professional awqāf management, however, is characterized by productive empowerment of awqāf and professionalism in management and includes management aspects,

human resources for awqāf supervisors (nazir), business partnership patterns, and forms of movable awqāf (Pertiwi et al. 2019, pp. 769–70).

## 4. Awqāf Bank

Due to its special nature, the question that constantly arises relates to the subject of awqāf categories, whether governmental agency, non-governmental organization, development finance agency or regular cooperation (Arnawut 2005, pp. 3–20). Many speak of the establishment of a conventional institution whose primary objective would be to provide favorable services for under-privileged or micro-financing of the poor in a bid to alleviate poverty (Bennett 1998, pp. 99–117). The fact remains that awqāf banks operate with the objective of micro-financing, and a special interest in the poor of the community (Abu Zahrah 1971, pp. 156–67). This may be conducted through various institutions, both private, non-governmental organizations and corporations.

The interesting question, however, pertains to who really should own the awqāf bank. This should not, however, define bank activities and transactions. The fundamental objective of awqāf bank remains alleviation of poverty, fulfilled though the issuance of affordable loans to the poor as described by Dreher (2006, pp. 769–88) while investing in the community's development, thus ensuring maximum benefit for the poor and those who can barely afford the cost of living. In this quest, they also need to ensure that the whole community, both rich and poor, is not infringed upon in the process of services provision. Awqāf banks owe the rich both a moral and spiritual responsibility of care, without disregarding their rights in the community (Renneboog et al. 2008, pp. 1723–42). When fulfilled, there would be no need to question the effectiveness of awqāf bank. What usually happens is that the generated income through awqāf bank-related activities are usually distributed among the under-privileged or recycled into the bank for services provision that would further help generate more income (Buttle 2007, pp. 1076–88). In the following section, we will address important issues such as the rationale of awqāf bank, its different advantages, legal framework, structure, and management as shown in Figure 1.

### 4.1. The Legality of Awqāf Bank under Conventional and Islamic Law

In current jurisdictions, there are various laws that regulate the formation and registration of companies, banks, and financial institutions (Bennett 1998, pp. 99–117). Banks may be registered as companies under the company act, with its own legal personality, rights, and obligations. Being its own legal personality, it would be considered separately from its directors and shareholders. In such capacity, it may hold land and other assets in its own name. The veil of incorporation and distinct personality can only be lifted in special circumstances (Zaman 1999, pp. 1–8). Banks may also be registered as cooperative with legal personality status like those of companies. In such a case, its members would have limited liability for they will not be liable for any debts incurred by the business. The law does not stipulate a structure of how the cooperative of awqāf banks should be formed, and thus the laws need to be amended to fit this purpose. The bank is then registered and incorporated in accordance with the banking and company laws and regulations put in place by the central bank of the Islamic jurisdiction (Dumith et al. 2011, pp. 24–28).

There are numerous legal statutes and sections that both define and regulate banks operations, including but not limited to, laws of Islamic banking, the Central Bank regulations, various conventional bank laws, and laws governing the cooperative and societies (Abu Zahrah 1971, pp. 156–67). However, of most importance are the Islamic Banking Laws since the awqāf bank is established in the Muslim states and the rules pertaining to the finances of the losses. This legal framework would for instance involves licensing, duties, financial requirements control of these financial institutions of such banks, business restrictions and powers to supervise (Ammar 2006, pp. 11–19). They are, however, stricter in their issuance of licenses as they depend on two major conditions (Arnawut 2005, pp. 3–20), alienation of their performance from any activity found to be inconsistent with the Islamic law and the mandatory prerequisite that the Articles of Association of bank formation must have the Shari'ah Advisory Council,

which advises the bank on how to operate in accordance with the Islamic laws (awqāf laws). The bank will generally be a new entity, and hence the need to enact new laws that regulate this institution or amend the existing laws to conform with the requirements of the law, especially when the bank is created (Barizah Abu Bakar et al. 2005).

### 4.2. The Rationality of Awqāf Bank

The main contribution of this study is made to the field of Islamic banking and finance. This research sheds light on the very nature, justification, and process of creating an awqaf bank, in the current economic and financial context where awqaf are highly dynamic and also effective in the betterment of communities, yet need to be re-structured into awqaf banks. This paper argues that such a transformation of awqaf funds to effective awqaf banks would better serve the socio-economic needs of communities and better enhance the awqaf revenues and sustain their future.

Some, however, argue that awqāf banks may operate like any other conventional bank, especially when it fundamentally operates on cash awqāf. Others hold the opposite view against the establishment of awqāf bank, when if, however, these are done away with, and the Islamic laws were a bit reasonable, then the validity of the cash awqāf could be extended to the awqāf bank. In the recent past, some well-to-do persons pooled their funds to jointly create Cash Awqāf to establish several private universities in the wake of the enactment of the Private University Act, 1992 (Sadeq 2002). Among others, there is one organization which is partially created by Cash Awqāf, namely, the Social Science Institute (SSI). It has three funds. First, an endowment fund which is a Cash Awqāf. The money is kept in the investment fund of an Islamic bank, which operates based on *mudarabah*. The profits are spent for fulfilling certain Islamic objectives laid down in the constitution of SSI. Secondly, the general fund, the profit of which is used to meet the operational cost of SSI activities. Thirdly, a poor-fund consisting of Zakah is paid by SSI, which is used to help the poor, especially poor students. In this way, Cash Awqāfs have added a new dimension to the activities of charity in Bangladesh (Sadeq 2002).

In view of both the basic functions of cash awqāf, the philanthropic nature of services provision would be beneficial to the poor (Amin et al. 2003, pp. 59–82) through its benevolence and investments initiation set for poverty alleviation, they then qualify to involve the awqāf bank into reconsideration despite the earlier opinions against its formation (Arnawut 2005, pp. 3–20). With attention to the different theories about cash awqāf of the Muslims adhering to the Hanafi and Shafi'i schools of the law, particularly those pertaining to investment projects and funds management, the awqāf bank may be accepted in those particular communities (Buttle 2007, pp. 1076–88).

It should be noted, however, that awqāf bank may do well on many transactions and microfinancing activities needed by the Muslim communities. If the awqāf bank can enforce such transactions, it then qualifies to be recognized as a valid bank under the Islamic law. For the bank to be admissible, both the positive Islamic views on the cash awqāf and the major benefits that accrue to alleviate poverty need to be taken into consideration. According to Cizakca (2004, pp. 40–59), the merger of cash awqāf and the bank is in the best interest of the bank being recognized both for its actions and use of cash awqāf.

Awqāf is used to save the capsizing capitalistic financial systems of the Islamic nations and communities and unknown to the Islamic communities (Arnawut 2005, pp. 3–20). When awqāf banks use cash awqāf, then the chances of success increase significantly. Still, the above situation in which the awqāf bank is rejected and at the same time accepted will be avoided since they will be seen and regarded as one. Another issue related to the acceptability of awqāf banks is that it has a nature according to which deposits are issued and from those earnings the bank gains profits (Roodman and Morduch 2014, pp. 583–604). This feature may be justified under the law by way of the [consideration of responsibility of awqāf managers (*nuzzar*) for initiating and also facilitating awqāf bank investments while carefully deciding on the potential investment and individuals who can use the funded money for capital. Funds managers should not only oversee the bank activities, but also diversify its options and seek investments out, and that which is not related to the bank's activities at all, provided it is in the prospect of gaining capital (Vakulabharanam 2005, pp. 971–97).

The bank is set to service cheap loans while generating income to create the loan structure appropriate for those less privileged at nominal interest rates (Siddiqi 2004, p. 81). At the onset of the bank formation, the founders may contribute towards the initial capital. Their capital, however, does not belong to the bank. Once the bank has earned enough profits to sustain itself, then the successive earnings would be used for the various community services. The bank is set to earn profits, which are used in the provision of cheap loans to the poor while developing the assets of the bank through demand fees levied on the loans and any other means (Roodman and Morduch 2014, pp. 583–604). The bank would be authorized to get capital and funds from the public since there is no provision that prevents such a type of capital acquisition.

### 4.3. The Structure and Management of Awqāf Bank

When formed, the institution will have the capacity to collect, distribute, recollect funds, and manage charitable and non-charitable finances (Barizah Abu Bakar et al. 2005). Regardless of its entity as corporate or otherwise, the bank may operate through provision of services in a manner that generates profits to the bank obtained through capital contribution from the non-awqāf members and profits generated through bank assets. The bank services will originate from the dormant assets and property and will be offered to the public borrowing at minimal charges to the poor and welfare organizations (Dunya 2002, pp. 57–82). The bank structure may take different forms, depending on the type of jurisdiction, laws and what those laws and rules stipulate (Ammar 2006, pp. 11–19). It can also assume diverse forms depending on the form of ownership and capital contribution; it can be a sole awqāf bank or cooperative entity on profit sharing between investors and awqāf.

Speaking of the bank assets management, the balance sheet is the core aspect of bank assets management for it entails all of the assets, liabilities, and capital, whether from awqāf deposits, members' donations or bank shareholders (Mishra 2006, pp. 1538–45). Those factors in the balance sheet provide the overall picture about the bank even without a prima facie glance at the profit and loss account. To ensure that there is no falter in the awqāf progress, the investment assets should be aligned with low-risk and very liquid objects to ensure immediate returns (Abu Zahrah 1971, pp. 156–67). The nominal capital deposits should also be subjected to low-risk and long-term objects to ensure that they are maintained in the bank as they represent more of a liability than an asset.

The success of awqāf banks is gauged according to the achievements of the awqāf fundamental objectives and the services and initiatives completed for the poor community rather than the generated capital amount. For example, the awqāf bank should also consider the amount of benevolent loans issued instead of just the awqāf bank's loan recovery rate (Ammar 2006, pp. 11–19). To effectively achieve this, there is a need for effective governance and funds management. Those funds include those donated to the awqāf institution through charitable means, those earned by the awqāf bank's service charges, or borrowed from other organizations. Those need to be managed in accordance with the Islamic laws and under diligent management in order to ensure that services provision to the poor communities is rather performed smoothly (Obaidullah 2007, p. 3).

However, there are exceptions which would allow the bank to opt for conventional banking methods, especially those related to short-term investments and highly liquid products. The administration of this exception requires managers to be cautious in order to avoid operation beyond the financial ability of the awqāf bank while ensuring that bank investments are those that are highly likely not only viable but also those operating within the requirements of the Islamic laws (Renneboog et al. 2008, pp. 1723–42). Funds investment must adhere to the Shari'ah laws. Still, the laws and regulations pertaining to the given bank jurisdiction are of paramount importance; different Islamic jurisdictions have closely related regulations, but which have specific differences however minimal they might be. They must be adhered to (Siddiqi 2004, p. 81).

Awqāf banks would usually have deposits and capital set aside for the services of the poor and for development activities in accordance with the wishes and commitments of both the endowers and donors. In awqāf, deposits can be used for the services provision and at the same time generate

income through minimal charges administered on the loans that are at the disposal of poor and less privileged (Young 2010, pp. 201–23). It accepts deposits from various sources and then invest them in diverse avenues and provision of disparate social services. Deposits, especially in situations of nominal deposits where these are bank liabilities (Abu Zahrah 1971, pp. 156–67). The use of savings and current accounts is a minimal risk but does not generate profit. Investment benefits on the share capital, which pertains to profit and loss sharing, can be used to generate profit returns. Those, for example, include features such as transaction deposits and investment accounts, special or otherwise (Zaman 1999, pp. 1–8).

Transactions deposits are generally not used for the generation of business income, rather used to complete the short liquidity balancing in the balance sheet and are not actively involved in the investments that generate income. They are therefore set for depositors' safekeeping but cannot gain profits. Those are based on the Islamic principles of object deposit (*wadi'a*) and benevolent lending (*qard hasan*), which would allow the bank to give some provision for its depositors. Investment accounts are generally based upon the principle of unrestricted investment (*mudaraba mutlaqah*). In such transactions, the awqāf bank becomes the party with the freedom to oversee the investment activities and invest in any place of interest (Amin et al. 2003, pp. 59–82). The bank may also accept those deposits merely based on the Islamic law of that particular jurisdiction. They must, however, adhere to the Shari'ah laws and rules of Islamic banking, which require strict management of resources and capital of the awqāf funding.

Awqāf banks may opt to use those funds to establish and further enhance awqāf properties development. Awqāf activities are valid if they are viable and continue to serve their original purposes. The special investment funds, on the other hand, do not possess any significant distinction from the investment account and how it generates income (Bennett 1998, pp. 99–117). It may accept any investment opportunities as stated in the *mudaraba*, which must also be in accordance with the Shari'ah laws. The particularly distinctive feature is the fact that such investments are issued to big investors who deposit funds for a particularly identified category or social purposes and activities. Those deposits are usually issued directly to the given objectives or purposes for which they have been originally issued. However, awqāf funds can be used for urgent yet essential priorities, like funding major agricultural projects or development of urban settings (Amin et al. 2003, pp. 59–82). They may assist the state to provide governmental services. Nonetheless, the bank should not act beyond its mandate and should ensure that its activities are generally for the good of the poor (Dreher 2006, pp. 769–88).

The sources for awqāf bank capital are largely obtained from charity activities in the awqāf systems such as zakat and *sadaqah* (Kuran 2005, pp. 593–615). In many situations, it may initiate earned profits distribution to investors and other parties. Minimum charges can be levied on issued funds to cater for the bank expenses and to take care of unexpected currency depreciation. Such initiatives are prudent for short-term circumstances as they bring immediate returns. The investments from all sources shall then be invested in a manner consistent with the Shari'ah and the Muslim laws pertaining to that particular jurisdiction (Buttle 2007, pp. 1076–88). Awqāf often delves in business banking since there is a need for continuous bank success through lending activities and other income generating activities. With good fund management, the bank can be a success (Abu Zahrah 1971, pp. 156–67). The Islamic laws, rules and regulations encourage that the investment projects should not only be put in place to ensure a profitable return, but they be inculcated to ensure maintenance and development of awqāf property.

Islamic law requires that an operation area decided by the awqāf bank is permitted by the Islamic banking regulations and the Islamic laws in general. However, the bank needs to ensure that there is a risk aversion and that there are adequate measures in place to offer the best capital protection, especially the nominal capital that is put for safekeeping, as they are usually more of a liability than an asset to the awqāf bank (Klugman 2009, pp. 13–46). To ensure this effectively, awqāf banks need to ensure that when dealing with the Islamic conventional banks in any form of transactions, they should choose exclusively low to medium risk investments. High-risk investments are expressly

discouraged in the awqāf systems and institutions (Grodach 2011, pp. 300–9). Having said that, one needs to draw attention to the important nature of Investment in period of high returns. All the assets purchased through awqāf investment returns do not form any part of the awqaf themselves. In addition, the conditions stipulated by the endower (waqif) with regards to the use of returns are binding. The surplus after allocation to beneficiaries may be invested. A good example for this would be the use of portfolio approach in Shari'ah-compliant sectors as well as use the diversification and other risk minimization tools for awqāf banks

Nevertheless, the effective revenues and funds management require the awqāf bank to run a yearly assessment of its profit and loss account so as to ensure that there are no loses running in the bank, and in case there are, then there is a need to operate adequate measures to figure out the problem and consequently propose new ideas to end such a trend while making further profits. Funds distribution should be conducted in the manner prescribed by the directors and depositors of the bank. Zakat, charities and awqāf donations of the bank may be used in the provision of micro-financing services for poor communities. The bank can be used to develop awqāf properties. This is usually not very easy since awqāf banks and institutions are usually charitable organizations. They are essentially set to service the unsecured loans; most of the poor people do not usually have securities needed for the loans in the conventional banks. Conventional banks do not issue loans to awqāf banks as they are too much of a risk (Dumith et al. 2011, pp. 24–28). The bank also majors in the provision of micro-financing business and other micro-financing activities, which largely run on profit sharing and benevolent lending (*qard hasan*). For this lending, the issued deposits would be distributed among the poor and needy, dedicated to the community's welfare. They are generally issued as bank micro-credit activities (Abu Zahrah 1971, pp. 156–67). The issued loans returns are also re-invested or re-issued for the needy and poor. Awqāf microfinancing is usually used to procure cash awqāf to facilitate trading with poor communities. A section of its profits would be granted to the needy, and in most cases, this goes to those who are below the poverty cut line (Roodman and Morduch 2014).

*4.4. Advantages of Awqāf Banks Over Other Awqāf Institutions*

Based on their fundamental nature and utility such as beneficiaries and interests, Shari'ah-compliant banks should be allowed to operate (Mishra 2006, pp. 1538–45). Essentially, the bank benefits beneficiaries and the awqāf. It can earn large funds and assets, both liquid and otherwise, which may be used for many other functions (Abu Zahrah 1971, pp. 156–67). Those can provide enough capital for awqāf and progressive development of both the dormant and active assets of awqāf. Similarly, they can be used to make awqāf bank break become independent from other conventional banks and government subsidies, and can sustain itself independently (Buttle 2007, pp. 1076–88). It can also generate new awqāf assets while preventing current ones from annihilation or dissolution. Similarly, it may prevent cash awqāf from misappropriation or misuse, as when the bank is put in place, all the books of accounts are placed in check to gauge the accountability of awqāf bank managers while ensuring that the awqāf does not lose property, but only continues to gain from the agencies that are not known as awqāf institutions..There is also the fact that in a country with federal laws like the United States of America, it can help facilitate a process in which awqāf institutions go easily through state and federal regulations (Amin et al. 2003, pp. 59–82).

It is interesting to note that through its funds, the bank would be able to fund diverse communities' socio-economic programs. The bank also has certain obligations towards its beneficiaries; this can only be viable if the bank generates some income. The beneficiaries can receive the bank services with their preferred conditions. In conjunction with other benefits, there will be a constant supply of liquid assets to the awqāf. Those assets play a vital role in the reviving of dormant properties and in retaining the productive properties of awqāf. When those are revived and maintained, then there would be a good prospect for an increase in the funds supply that would in turn enable provisions for poor beneficiaries (Zaman 1999, pp. 1–8). At this stage, trustees experience difficulty in their operations because awqāf banks tend to have fixed assets which can only be used to gain profits by way of cultivation. Those assets

may be insufficient to fund new investments. Financing those development schemes may be extremely difficult as they are unsecured loans, which would not usually suffice as options in conventional banks. Those institutions that are almost failing can be refinanced through awqāf bank, and hence would have aided both the institutions and its beneficiaries (Amin et al. 2003, pp. 59–82). For the country of Bahrain for example, Sarea argues that there is substantial support for the establishment of a new waqf bank. He maintained that strong support manifests for the new waqf bank to be structured as an investment and development bank operating, domestically and internationally, as an independent corporation in terms of legal status (Sarea 2019).

## 5. Conclusions

As stated earlier, the awqāf bank would drive the economy towards improved socio-economic standards, thus effectively lifting the lower bar of income per capita as well as unpopulated crowds from poverty. Such results are expected to happen according to the perspective of awqāf funds investment and development which provide affordable financing to lower income demographics, helping them break the cycle of poverty. However, Awqāf bank interested beneficiaries may be concerned with the mismanagement, misuse, and misconduct of the awqāf managers and administrators, as well as their uncertainty of their profitable investment to grow the bank's income. The bank should solicit funds from the public, Islamic corporations, governments, Islamic banks, awqāf capital, awqāf reserves, zakat ministries and public charities. Awqāf bank maintains its raison d'être of perpetuity for the poor and needy through provision of free interest yet affordable financing and full banking services. Awqāf bank should be an independent statutory body, financially based on equity participation such as *mudaraba* or *musharaka*. Awqāf capital and awqāf reserves should be invested in the protection of its capital and generation of more income. The bank should be Shari'ah-compliant, with its funds' investments based on participants' equity. Its use of capital of charitable funds should be zakat and charities (*sadaqat*).

This study highlights the legitimate need for Shari'ah-compliant awqāf banks, which in addition to providing basic services for its beneficiaries and for society at large, would also manage investments and awqāf funds in such a way that would make impactful contributions to overall national development and economic growth. This research would be of high relevance to experts, practitioners, financial managers, regulators, and policy makers in the fields of awqāf, banking and finance. Future lines of research should embrace both theoretical and empirical studies, and would address several issues related to the concept methodology and indexes of Sharia compliance, the role of the *maqasid* in the management of awqāf financing and banking, awqāf funds' investments in contemporary societies, in addition to evaluation studies of models of awqāf banks.

However, it is worth remembering that in spite of the anticipated advantages of a Shari'ah-compliant awqāf bank in poverty eradication, improving human welfare and development in general, it essentially belongs to a much broader perspective of awqāf, social welfare and civilizational development. The awqāf, as a voluntary and irrevocable dedication of Muslim wealth to God, continues for a long period, toward the development of many critical sectors of society, including the most needed areas and welfare of most aspects of community life and people. The awqāf also provides essential services such as food, education, health, accommodation, and general infrastructure. This only implies that the proposed awqāf bank is not expected to function alone nor in a vacuum of social change, but rather only complements the community's ethical and religious commitment to brotherhood, solidarity, cooperation, and equality.

**Author Contributions:** Conceptualization, H.G. and T.T.; methodology, B.B.; formal analysis, F.J.; writing—original draft preparation, H.G.; writing—review and editing, B.B.; supervision, T.T.; All authors have read and agreed to the published version of the manuscript.

**Funding:** This research received no external funding

**Conflicts of Interest:** The authors declare no conflict of interest.

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
