# Peer review of "The Need for Shari’ah-Compliant Awqāf Banks"

_jrfm, doi:10.3390/jrfm13040076_

Round 1

Reviewer 1 Report

Dear Author(s),

The paper "Towards Shari‘ah Compliant Waqf Bank" is a good paper with interesting topic that explores prospective of the Waqf bank. I would suggest it to be published where it has an original idea and clear explanation for the content.

The authors has explored the phenomenon in a clear way and the proposal of the Waqf bank with it pros and cons and the way for improvement. 

I recommend your paper for publication.

Author Response

Thanks

Reviewer 2 Report

First of all thank you for the opportunity to read your work. Despite the topic seems interesting it has several issue. The main problems are the contributions and the study approach adopted.

Introduction. In the introduction it is not clear the aim of the work. In fact it looks like an analysis of literature, this makes the positioning of the paper relatively clear, however: what are the contributions of this work? Which is the research gap? Why this study is relevant?  Please, in your introduction, underline these aspects in detail.

Methodology. The methodology applied is unclear. If the objective is to understand the literature on this topic I suggest to apply a systematic literature review. In the current state the work presents only a list of characteristics and is not sufficient to make a contribution to the literature on the subject.

Conclusion. The conclusions are not enough in this state. I recommend to insert both theoretical and managerial contributions. Also please add the research limits and future research lines on the topic.

Furthemore, provide a profession proof reading service to avoid typo errors.

Author Response

Dear Reviewer

We have indeed benefited from your important comments and have taken them all very seriously.

Please check the new submitted version that shows the integration of all changes based on your advices and recommendations.

Best

Reviewer 3 Report

Its my pleasure to read the paper. It is a good paper directing to establish and operate Waqf Bank which would expect to benefit the humanity (not only Muslims) a lot.

The paper could be significantly improved by - 

  1. Tabular/ graphical presentation of advantages, Legal framework, Management Structure.
  2. Need thorough English Check, there are some major issues with writing.
  3. Conclusion should provide a thorough review of the paper, its contributions to the literature and policy and further research directions.  

Author Response

(The authors gave the same response as above.)

Reviewer 4 Report

The authors of the research paper Towards Shari‘ah Compliant Waqf Bank addresses a relevant theme on the development and provision of basic services, agreeing to identify global solutions to reduce inequalities.

The concepts, the specialized literature and the bibliographic sources are suitably selected by the research authors, and especially on the research topic "Waqf Bank". Moreover, authors such as Bennett who address the micro-financing of the poor in an attempt to alleviate poverty, or Dreher who defines the fundamental objective of the Waqf Bank as poverty alleviation, through cheaper, but accessible loans to the poor, are mentioned clearly and argued by the authors of the paper.

The research methodology corresponds to the type of empirical research based mainly on analytical documentation, as is the case of this reviewed work. However, the methods, models and instruments specific to the classical research are limited presented, which is why the specific instrument of this work is the case documentation.

The research results are based on the direct mention of the links between Waqf Institutions and Social Banks, The Issue of Cash Waqf, Waqf Bank (Rationale for Waqf Bank, Advantages of Waqf Bank, Legal framework of Waqf Banks, Structure and management of Waqf Bank,) and duly documented by the authors of the research through appropriate bibliographic reference sources. However, to know the personal scientific contribution of the research authors, we suggest a distinct mention of the results of the scientific research. Especially, due to the fact that the authors of the research mention that this type of financing ”would significantly help the economy in a constant and sustainable growth; efficiently lifting the lower bar of income per capita and removing the unpopulated crowds from poverty ”, an extremely important aspect that can be disseminated globally by publishing such works.
The conclusions are adequately presented by the research authors, namely that the Waqf Bank is vitally important for the poor and needy, namely the provision of free loans and services, but also affordable prices and complete banking services. However, we suggest in addition to the application (practical) aspects mentioned by the research authors to be mentioned and scientific research (personal contributions) aspects being very important both as a result of the subject addressed and for the direct contribution to the specialized literature.

With these additional completions regarding results and conclusions, we propose after reviewing the acceptance for re-analysis for publication the work, the topic being very important in the context of global inequalities.

Author Response

(The authors gave the same response as above.)

Round 2

Reviewer 2 Report

Paper has been improved from previous version.

Authors  have implemented all my suggestions. I think that the paper can be published in the current form.

Good luck for your future research.

Author Response

First we would like to the thank the Editor immensely for his very constructive comments and generous correction. Truly, we found them to be extremely useful.
We have tried our very best with the allocated time to address most of the points including for example, review of language, extended discussion of Cash Waqf for which we had to resort to more sources, review of the section of social bank, governance structure, further highlight of the advantages of cash waqf, and much more.
I hope you would find this satisfactory.

Authors

Reviewer 4 Report

With these improvements made by the authors of the research within the paper, I also suggest a careful reading of the paper (checking the English language) and we propose the acceptance for publication of the paper and congratulate on the relevant topic of the research.

Author Response

(The authors gave the same response as above.)
